# Recent Advances in C-S-H Nucleation Seeding for Improving Cement Performances

**DOI:** 10.3390/ma16041462

**Published:** 2023-02-09

**Authors:** Ana Cuesta, Alejandro Morales-Cantero, Angeles G. De la Torre, Miguel A. G. Aranda

**Affiliations:** Departamento de Química Inorgánica, Cristalografía y Mineralogía, Universidad de Málaga, 29071 Málaga, Spain

**Keywords:** accelerators, admixtures, C-S-H nanoseeds, ettringite, microstructure

## Abstract

Reducing cement CO_2_ footprint is a societal need. This is being achieved mainly by replacing an increasing amount of Portland clinker by supplementary cementitious materials. However, this comes at a price: lower mechanical strengths at early ages due to slow pozzolanic reaction(s). This is being addressed by using accelerator admixtures. In this context, calcium silicate hydrate nucleation seeding seems to have a promising future, as it can accelerate cement and pozzolanic reactions at early ages, optimising their microstructures, without compromising late strength and durability performances. In fact, these features could even be improved. Moreover, other uses are low temperature concreting, precasting, shotconcrete, etc. Here, we focus on reviewing recent reports on calcium silicate hydrate seeding using commercially available admixtures. Current knowledge on the consequences of nucleation seeding on hydration reactions and on early and late mechanical strengths is discussed. It is noted that other features, in addition to the classic alite hydration acceleration, are covered here including the enhanced ettringite precipitation and the very efficient porosity refinement, which take place in the seeded binders. Finally, because the seeded binders seem to be denser, durability properties could also be enhanced although this remains to be properly established.

## 1. Introduction

A major objective for PC production is to achieve carbon neutrality by 2050. This ambitious goal requires that several research avenues to successfully converge. One of these approaches is to decrease binder CO_2_ footprints. A leading study [1] identified PC clinker replacement by SCMs [2,3] as the most promising strategy today and in the near future. This approach currently has the lowest performance and economic impacts. However, the key current drawback of low-carbon cements is their low hydration rate at early ages, leading to poor mechanical performances at three days of hydration, or earlier. Therefore, there is an obvious need to increase the early age reactivity of PC-SCMs blends [2,3,4,5], which is shared by other low carbon cement proposals [6,7,8].

There are different approaches to activate/accelerate early age hydration reactions (dissolution and precipitation processes) of PCs [9], one possibility being the addition of chemical admixtures. The chemical acceleration of PC hydration by admixtures has been recently reviewed. Different setting retarders, accelerators, and water reducing dispersants were classified with emphasis on the current understanding of interactions with cement chemistry [10]. Moreover, a selection of accelerators specially used in shotcrete have also been revised and their effects on hydration, microstructure and properties were analysed [11]. Very recently, admixtures to be used in normal concrete have been classified in chloride-based accelerators and in non-chloride accelerators, putting emphasis in their effects for durability [12]. The accelerators for cast and sprayed concretes were also revised highlighting the influence of the dosage on the aluminate and silicate hydration reactions [13]. It is well known that some accelerators, such as calcium salts (CaCl_2_, Ca(NO_3_)_2_, Ca(NO_2_)_2_, Ca(SCN)_2_), sodium metasilicate (Na_2_SiO_3_), or sodium aluminate (NaAl(OH)_4_), promote the early hydration of cements [11,13,14,15,16]. However, it must be also kept in mind that some chemical admixtures may present durability problems or they can reduce the mechanical strengths at 28 days or later [13,17]. Moreover, alkanolamines are widely used as grinding aids and they are now also being employed as accelerating admixtures with variable performances depending upon the chemical formula of the alkanolamine and its dosage. It was reported that TEA accelerates the hydration of mainly C_3_A and C_4_AF, and it slows C_3_S hydration [18,19]. TIPA mainly affects to C_4_AF hydration [18,20]. Moreover, it was also proved that TEA and TIPA enhance the overall hydration degree when they are used in SCM-containing cements [21], including LC^3^ [20]. DEIPA and EDITA lead to C_3_A and C_4_AF hydration acceleration promoting more AFm formation [18,22]. Finally, THEED and THPED were also studied in this context [18].

One approach which does not seem to show adverse consequences, neither in late mechanical strengths nor in durability performances, is nucleation seeding with water-based stabilised suspensions of C-S-H nanoparticles [23,24]. In short, this approach is known as C-S-H seeding. The use of C-S-H seeding for accelerating cement hydration was reviewed [25,26]. Hence, we restrict this work to the most recent advances with just some references to previous key works. It is explicitly stated that this work does not cover nucleation seeding with other type of nanoparticles, i.e., SiO_2_ [27,28,29,30,31,32], Al_2_O_3_ [27,29], TiO_2_ [29,30,31], and CaCO_3_ [30,32]. C-S-H seeding of cements is a fast-growing field which is expected to gain momentum because of the need to accelerate the hydration of low-carbon cements and simultaneously to improve their durability performances.

The importance of C-S-H nucleation seeding is, therefore, due to its ability to accelerate the hydration of low-carbon cements at early ages without affecting their durability performances at later ages. This is reflected in higher mechanical strengths at early ages. Moreover, C-S-H seeding is also important for concreting at low temperatures as hydration acceleration is required.

## 2. Brief Discussion on C-S-H Nucleation Seeding Mechanism as It Was Known before 2020

C-S-H nanoparticle seeding is known to have two main effects on cement hydration [25,26]. On the one hand, these nanoparticles modify the pore solution ion concentrations (e.g., Ca^2+^, Al^3+^, Na^+^, K^+^, SO_4_^2−^, etc.), as these ions are effectively adsorbed on the seeds because their very large specific surfaces. The reversible adsorption of ions from the pore solution, in turn, has three main consequences: (i) the changes in the ions concentration in the pore solution creates local concentration gradients, which could alter the dissolution pathways in cementitious systems, (ii) the C-S-H growth mechanism(s) could be changed due to the adsorbed species, and (iii) most importantly, the changes in SO_4_^2−^ and Al^3+^ concentrations in pore solution indirectly affect, it may even delay, alite dissolution. The roles of aluminate [19,33,34,35,36] and sulphate [37,38] species in alite hydration rate is being very actively researched. These effects are not directly related to the nucleation process.

On the other hand, C-S-H seeding supplies additional nucleation sites, which are very suitable because of their low interfacial energy. There are two very different consequences from this nucleation process. First, it may physically accelerate calcium silicate hydration, also known as the filler effect. This is due to the coupling between C-S-H precipitation and alite dissolution. Second, synchrotron X-ray diffraction tomography directly showed that C-S-H seeding partly moves the C-S-H gel nucleation and growth away from the dissolving clinker particles, enhancing secondary heterogeneous nucleation in the capillary porosity [39]. The rearrangement of the C-S-H gel from the surfaces of alite particles to the pore space leads to a more homogeneous C-S-H distribution in the bulk of the paste [40], which in turn decreases the porosity, improving the mechanical strengths, and reducing the permeability of the final binder [41]. This can be indirectly evidenced by lower porosities and smaller threshold pore entry sizes, as measured by MIP. Figure 1 schematically summarizes the influence of C-S-H nucleation seeding in the hydration of PC.

C-S-H seeding is quite different in cements, where there are many ions in the pore solution (including sulphates and aluminates), compared to pure alite pastes, where the pore solution does not have these ions. For instance, crystalline afwillite, Ca_3_[SiO_4_][SiO_2_(OH)_2_]·2H_2_O, is known to accelerate the hydration of C_3_S pastes, but not the hydration of PCs [42]. Moreover, by using the micro-reactor technology, C-S-H seeding accelerated C_3_S hydration, but did not accelerate, in fact it slowed, PC-slag hydration in alkaline media [43,44]. These observations highlight the difficulty to extend the results of C_3_S hydration acceleration by nucleation seeding to PC activation using the same admixture. Due to this critical difference, the present review does not discuss on the consequences of C-S-H seeding for alite pastes.

In this context, the influence of PC mineralogy [45] and cement fineness in the C-S-H seeding performances have been published. For PCs with larger surface area, acceleration by C-S-H seeding is comparatively lower [45].

The novelty of this work resides on the discussion of the consequences of C-S-H nucleation seeding beyond the accelerating effect on alite hydration at very early ages. Specifically, several other outcomes are emerging from the analysis of the reports in recent years, including: (i) the importance of faster ettringite crystallisation; (ii) the possible slower alite dissolution rate after approximately one day because the higher concentration of aluminate species in the pore solution; and, chiefly, (iii) the lower porosities and threshold pore entry values of the seeded pastes.

## 3. Review Objective and Methodology

The main objective of the present review is to gather the latest data concerning the consequences of C-S-H nucleation seeding using commercial admixtures. This is done with the intention to develop a better mechanistic understanding that could results in further improvements. It must be noted that a wide range of techniques are being currently used to characterise the seeded products, and, hence, a better mechanistic understanding can now be developed. C-S-H nucleation seeding is especially suitable for low-carbon binders and for concreting at low temperatures. In these cases, the slow hydration rates must be boosted to have competitive mechanical performances. Moreover, because denser binders are produced, all works related to durability characterization are discussed.

The employed methodology and the structure of the review is presented next. On the one hand, two tables gather the latest works using commercial admixtures and the relative improvement of the mechanical performances at early and later ages. Additionally, the structure of this work is also detailed. Section 4 reports the data collection features. Then, key Section 5.1 thoroughly describes the works employing commercial admixtures. The corresponding works developing and using laboratory-prepared C-S-H seeds are analysed, in less detail, in Section 5.2. Then, the current understanding of the processes involved in the activation of cement hydration by C-S-H nucleation seeding are reviewed, Section 6. This is divided in three subsections: (i) the acceleration of cement hydration at early ages; (ii) the improvement of mechanical strengths at early and later ages; and (iii) the improvement of durability performances at later ages. Finally, Section 7 gives an overall picture, as currently understood, of the mechanism at stake in C-S-H nucleation seeding. The review ends with a section of conclusions, where also the most important future research needs are highlighted.

## 4. Data Collection and Categorizing

In this section, we briefly discuss the data collection procedure and an analysis of the referenced works. Most of the investigations cited here are papers from journals, amounting 91% of all references, i.e., 112 out of 123. It is noted that this work references four book chapters, five proceedings, and two patents. The investigations have been mainly obtained from the three available databases: (i) Google Scholar; (ii) Scopus; and (iii) Web of Science. The searches were based on keywords “C-S-H” or “CSH”, and “seeds”, “seeding”, or “nanoseeding”. Moreover, the introductions of the resulting investigations led to some works which were not identified in the initial searches.

As a review on C-S-H seeding was published in 2018 [25], a time analysis seems adequate. It is noted that 82, out of the 123 references gathered here, have been published in 2019–2023. Therefore, at least 67% of the references were not covered in the previous review. Moreover, it is remarked that in 2021, 32 papers where published dealing with C-S-H seeding. This is the last year we consider that it has been fully covered here, as the searches for this review were conducted in November 2022. The corresponding publications in 2017, 2018, and 2019 were 5, 7, and 9, respectively. This four-fold increase in four years shows the growing importance and momentum of ‘C-S-H seeding’.

## 5. C-S-H Nucleation Seeds

In this review, we distinguish commercially available C-S-H nucleation seeding admixtures from C-S-H nanoparticles prepared in a given laboratory. More attention is paid to the studies employing commercial admixtures. The reasons behind this choice are threefold. First, the availability of commercial admixtures allows us to use larger quantities for enhancing the hydration of mortars and concretes. Second, and quite related, this readiness also permits the study of the acceleration of PC-SCMs blends and the possible improvement in durability performances. This type of studies requires large amounts of homogeneous admixtures, which are generally not available from laboratory preparations. Finally, the commercial admixtures enable studies from different research groups for similar binders and/or applications. The availability of a large body of research increases confidence if similar results are obtained. The T^3^ principle, ‘tried, tested, and trusted’ is key for the widespread adoption of new approaches in the conservative construction and building materials field.

Conversely, laboratory prepared C-S-H nucleation seeds mainly allow the investigations for that particular research group. This could be interesting for developing alternative synthetic procedures and/or different final products, but it comes to a price. It should be noted that the reproducibility of these syntheses could be problematic as several nuances apply: (i) average Ca-Si ratio and dispersion in the final solids; (ii) specific surface, size and shape of the nanoparticles; (iii) nanoparticle size distribution; (iv) stabilization features of the nanoparticles within the suspensions; and (v) possible aging effects.

### 5.1. Commercial C-S-H Nucleation Seed Admixtures

As discussed above, the addition of pre-formed calcium–silicate–hydrate nanoparticles enhance the development of the C-S-H gel produced from the hydration of the silicate phases in PCs. This was known, at least, since early 2000 [46,47]. A breakthrough came with two world patents in the years 2010 and 2011 awarded to Construction Research & Technology GmbH, a subsidiary of BASF [48,49]. These patents dealt with the use of C-S-H stabilised suspensions, employing plasticizers/polymers, to accelerate/enhance the hydration of cementitious materials. This started the Master X-Seed family of concrete accelerators based on C-S-H nucleation seeding technology. As soon as 2011, the first paper employing a commercial C-S-H based admixture was reported [50]. In this work, X-Seed 100, with 7 wt% of solid content (stabilised C-S-H nanoparticles) commercialised at that time by BASF, was compared to other accelerating admixtures. Several other papers describing the PC hydration acceleration features of Master X-Seed 100 were subsequently reported [45,51,52,53]. The Master X-Seed family of products are now produced and commercialised by Master Builders Solutions.

In the following paragraphs, we list and clarify the currently available commercial admixtures, as known by the authors. Research papers using these admixtures are listed in Table 1, where some key features are also highlighted. Many papers do not report the solid content of the admixtures; which should be avoided. The solid contents must be given to have a better understanding of the results and to frame the outputs in the C-S-H nucleation seeding landscape. In some cases, the solid contents are determined by constant weight at 105 °C. However, C-S-H nanoparticles may have lost weight at that temperature. We strongly advise determination of the solid content from the constant weight at 40–45 °C, acknowledging that it can take between 3 to 7 days.

The Master X-Seed family of stabilised suspensions are fabricated and commercialised by Master Builders Solutions: (i) Master X-Seed 100 currently contains a concentrated suspension of C-S-H nanoparticles, ≈23 wt%, stabilised with chemicals including superplasticisers; (ii) Master X-Seed 55 is a related suspension commercialised in North America; (iii) Master X-Seed 1500 is commercialised in Australia; and (iv) Master X-Seed 120 is commercialised in Germany—in this case, it is nitrate-free. Moreover, new developments have resulted in (v) Master X-Seed 130, which has a slightly higher overall solid content, ≈28 wt%, and it also includes alkanolamines. This admixture is specially tailored to enhance very early age mechanical strengths for precast industries and LT concreting. Finally, (vi) Master X-Seed STE 53 is related to Master X-Seed 130, as it also has a solid content of ≈28 wt%, but this admixture is aimed at improving both early and late age mechanical strengths, particularly adequate for SCMs-containing low-carbon cements.

The HyCon^®^ S family of products are solid powders produced by BASF Construction Additives GmbH (Trostberg, Germany). These solid powders are based on C-S-H nanoparticles dispersed by superplasticizers. These dry products obviously require homogenous dispersion in the pastes/mortars/concretes.

In China, Shanghai Sunrise Polymer Materials Co., Ltd. (Shanghai, China) commercialises the product VIVID-300(CN). According to the information provided in its web page, VIVID-300(CN) is a stabilised suspension, 21 wt% of solid content, which is especially recommended for precast concrete.

Mapefast Ultra is a water suspension containing hydrated silicate nanoparticles produced by Mapei (Milan, Italy). This admixture is also particularly suitable for precast concrete.

Finally, the authors are aware of one report dealing with a C-S-H suspension admixture produced by a company reported as Changan Construction Material Co., Ltd, see bottom of Table 1. However, a web search in English did not give the corresponding admixture. Hence, no further discussion is elaborated here.

Moreover, there are reports employing C-S-H based commercial admixtures, but without stating the origin/details of such admixtures. These works cannot be reproduced and they are cited here just as examples of reporting practices that should be improved [54,55,56,57]. An intermediate case is the studies by Golewski and Szostak [58,59], where they indicated that the admixture was Master X-Seed, but the actual type was not detailed.

**Table 1 materials-16-01462-t001:** Summary of research works employing commercial admixtures based on C-S-H seeding.

Product ^#^ Name [Solid Content, wt%]/Year	Dosage ^@^ (wt%)	Binder	w/b	Selected Details	Ref
Master X-Seed 100 * [24]/2016	0.04–3.0 *0.16–12.3*	CEM I 52.5N	0.45	Calorimetry study comparing the hastening performances of 12 accelerator admixtures	[15]
Master X-Seed 100 [-]/2018	- *3.7*	CEM II/A-LL 42.5	0.50	Comparing 3 admixtures for plastic shrinkage cracking mitigation in concretes	[60]
Master X-Seed 100 [-]/2018	- *4.0*	CEM I 32.5R—FA	0.36, 0.40	t-dependent (4 h to 28 d) compressive strength improvement for concretes	[61]
Master X-Seed 100 [-]/2019	- *2.0*	CEM V/A	0.30, 0.50	t-dependent (12 h to 28 d) compressive strength improvement for mortars with SCMs	[62]
Master X-Seed 100 [-]/2020	- *0.5–5.0*	PC G-type OWC	0.44	Acceleration studies for pastes (calorimetry and UPV) at 25 °C, 40 °C, and 60 °C	[63]
Master X-Seed 100 [-]/2020	- *4.0*	CEM I 32.5R—FA	0.30	t-dependent (8 h to 28 d) compressive strength improvement for pastes with FA	[64]
Master X-Seed 100 [-]/2020	- *2.0*	CEM I 42.5R	0.29	t-dependent (6 h to 28 d) compressive strength increase for concretes	[65]
Master X-Seed 100 [-]/2021	- *1.2*	PC M 400-D0	0.26	t-dependent (2 h to 14 h) strength evolution for concretes	[66]
Master X-Seed 100 [-]/2021	- *4.0*	CEM I 32.5R—FA	0.26	Mechanical strength values from [64] plus new rheological and SEM data	[67]
Master X-Seed 55 [-]/2021	0.07 *1.0*	PC A3000—MK	0.40	PC with low grade MK. Compressive strength improvement for pastes, 1–28 d	[40]
Master X-Seed 120 [0.14]“/2020	- *1.0–3.0*	CEM I 42.5N—CC	0.50	Accelerators: nitrate-free C-S-H seeds and micro-limestone. Complete calorimetric study	[68]
Master X-Seed 1500 [23] ^=^/2021	0.12–0.35 *0.5–1.5*	PC—FA	≈0.26	Acceleration of high-volume FA concretes, 1–14 d. Environmental impact calculations	[69]
Master X-Seed 1500 [23] ^=^/2021	0.12–0.35 *0.5–1.5*	PC—FA	≈0.25	Strength data at 14 d in [69] are replaced by values at 28 d. Paste microstructures studied in detail	[70]
Master X-Seed 130 [≈28]/2022	0.60 *2.0*	CEM I 42.5R & BRC	0.50	In situ SXRPD study also using XS100 and TIPA as reference admixtures. Comparison of calorimetries for pastes with UPV for mortars	[71]
Master X-Seed 130 [24]/2022	0.48 *2.0*	CEM I 52.5R	0.35	GBFS as fine aggregate, substitution 25–100%. LT curing. Frost and acid attack resistances	[72]
Master X-Seed 130 [24]/2022	0.12–1.44 *0.5–6.0*	CEM I 52.5R	0.23, 0.27	LT curing. Antifreeze admixture. Compressive strengths. Some durability data	[73]
Master X-Seed 130 [24]/2022	0.48, 1.2 *2.0, 5.0*	CEM I 52.5R—GGBFS	0.27	T = 0 °C curing. Compressive strength and frost resistance improvements by C-S-H seeding	[74]
Master X-Seed 130 [23]/2022	2.3 *10*	CEM I 52.5R—GGBFS	0.42	NaNO_3_ as antifreeze admixture. Curing at −15 °C. UPV data. Compressive strength data	[75]
Master X-Seed 130 [-]/2022	- *1.5, 2.0*	CEM I 52.5R—SF	0.21	Comparison of four accelerators for 3D printed concretes	[76]
Master X-Seed 130 [≈28]/2023	- *2.0*	BRC-LC^3^	0.40	Improvement of mechanical performances in mortars. RQPA, TA, and MIP data for pastes	[77]
Master X-Seed STE53 [≈28]/2022	0.60 *2.0*	CEM I 42.5R & BRC	0.40, 0.50	RQPA, TA and MIP data at 1 d, 7 d, and 28 d, for pastes, respectively. Compressive strengths for mortars. It also contains data for XS130	[78]
HyCon^®^ S ^$^ 3200 F [100]/2020	2.0 *2.0*	CEM I 52.5N	0.30, 0.35	Impact of seeding in autogenous shrinkage. Use of SAP. MIP and X-ray microtomography	[79]
HyCon^®^ S ^$^ 7042 F [100]/2022	1.5, 3.0 *1.5, 3.0*	CEM I 52.5R—GGBFS	0.40, 0.50	Strength enhancement up to 180 d. Degree of hydration are reported. RQPA and MIP data	[80]
C-S-H from Sunrise ^%^ [99.8]/2020	0.5-2.0 *0.5–2.0*	P^.^I 42.5	0.50	Improving setting and early strengths for pastes and mortars containing EVA copolymer	[81]
C-S-H from Sunrise ^%^ [-]/2021	- *0.5, 1.0*	P·I 42.5—FA	0.40	Synergic action of C-S-H seeding and TEA addition in a FA-containing blend	[82]
C-S-H from Sunrise ^%^ [-]/2021	- *1.0*	PC G-type OWC—FA	0.44	Curing at 10 °C. Several types of additions	[83]
C-S-H from Sunrise ^%^ [-]/2021	- *0.5–2.0*	P·II 42.5	0.40	C-S-H, TIPA and NaCl additions. Chloride binding capacity measurements	[84]
C-S-H from Sunrise ^%^ [-]/2021	- *0.5-2.0*	P·I 42.5	0.50	Combined use of C-S-H seeding and sodium sulphate addition for hydration acceleration	[85]
C-S-H from Sunrise ^%^ [-]/2021	- *0.5–2.0*	P·I 42.5	0.50	Combined use of EVA, TEA, and C-S-H seeding in PC hydration	[86]
C-S-H from Sunrise ^%^ [99.8]/2020	0.5–2.0 *0.5–2.0*	P·I 42.5	0.50	Study of the synergistic effect of TIPA and C-S-H seeding on PC hydration	[87]
C-S-H from Sunrise ^%^ [20]/2021	0.8–1.2 *4.0–6.0*	Four P·I 42.5 cements	0.42–0.48	LT curing, 5–20 °C, of shotcrete also containing alkali free liquid accelerator	[88]
Mapefast Ultra ^&^ [23]/2022	-	-	-	Cu doped C-S-H seeds studied by SAXS and Total X-ray scattering. No works in mortars yet	[89]
Nano C-S-H ^?^ [20]/2021	0.25–0.75 *1.25–3.75*	PC 42.5	0.36	Autogenous shrinkage. Compressive strength data for pastes. MIP data	[90]

^#^ Master X-Seed, VIVID, and Mapefast Ultra are stabilised suspensions. HyCon^®^ S is a dry powder. In square brackets are given the dry residues (wt%), when given in the publication. ^@^ The dosages (bwb) are referred to the dry residue. Dosages referred to the commercial admixtures are given in italics. * Master X-Seed family (stabilised suspensions) is provided by Master Builders Solutions. “ The authors reported in the publication a liquid suspension with 0.138 wt% of solid content, which likely contains an error as this C-S-H content is too low. ^=^ The authors in references [69,70] wrongly quoted the solid content of Master X-Seed 1500 as 77 wt%. This is the approximate water content, the solid content being ≈23 wt%. This error is corrected in the table. ^$^ HyCon^®^ S family (dry powders with nanoparticles dispersed by polycarboxylates) is provided by BASF Construction Additives. ^%^ Shanghai Sunrise Polymer Materials commercialises VIVID-300(CN), which is a stabilised suspension with ≈21 wt% of solid content. ^&^ Mapefast Ultra (stabilised suspension) is provided by MAPEI S.p.A. ^?^ The authors of the publication stated that the seeding admixture was provided by Changan Construction Material. This product could not be verified by the authors of this review through a web search in English.

### 5.2. Laboratory Synthesis of C-S-H Nucleation Seeds

The preparation of non-stabilised C-S-H seeds has been reported based on different chemical reactions [91,92,93,94,95,96,97,98,99,100]. These works are not discussed here as the PC hydration activation performances are generally poor compared to stabilised C-S-H seeds. This was elegantly demonstrated by comparing the cement activation performances of C-S-H seeds identically prepared, but with and without PCE stabilization [101]. Moreover, time dependent results can be obtained for non-stabilised seeds as ageing could lead to a degradation of the performances.

There are also reports stating the synthesis of stabilised C-S-H seeds without any detail about the employed dispersing agent. This work is also not discussed here [102].

#### 5.2.1. Stabilised C-S-H Seed Composites

In 2017, Plank’s group reported the laboratory synthesis of stabilised C-S-H/PCE composites by co-precipitation of Na_2_SiO_3_ and Ca(NO_3_)_2_ solutions. In this study, IPEG-PCE based superplasticizer was employed as stabilizer [103]. Activation performances were tested in mortars using a PC 32.5, w/c = 0.44 and a dosage of 0.35% bwc, referring to the solid composite content. Seeds prepared at pH ≈ 12 gave the best results. Subsequently, the same authors used the admixture to activate a CEM II/B-V blend cement containing 35 wt% of FA [104]. The hydration activation of mortars and concretes were studied with w/c = 0.41 and dosages of 0.8%, 1.4%, and 2.0% bwc, respectively, referring to the solid content. The admixture accelerates the hydration of the clinker silicate phases, and substantially promotes the FA pozzolanic reaction. Later, related composites were stabilised with b-naphthalenesulfonate-formaldehyde and tested with two PCs, i.e., 42.5 and 52.5 [105]. Finally, three different C-S-H/polymer composites were also prepared by co-precipitation. The stabilisation was ensured by employing: (i) acetone-formaldehyde-sulphite, (ii) a melamine-formaldehyde-sulphite, and (iii) a sodium β-naphthalenesulfonate-formaldehyde. The accelerating performances were tested on a FA-limestone blended, CEM II, cement. The admixture dosages were either 1.3%, 2.0%, or 3.0% bwc, respectively, and the w/c ratio was 0.50 [106].

C-S-H/PCE composite seeds have been generally fabricated by co-precipitation of Na_2_SiO_3_ and Ca(NO_3_)_2_ in the presence of stabiliser polymers. For instance, C-S-H/PCE and C-S-H/PSE composite seeds were stabilised with either HPEG-type PCE or by a polysulfonate derivative [107]. The activation of pastes was studied only by isothermal calorimetry. The pastes were prepared with a PC 42.5, w/c = 0.40 and variable dosage from 0.0 to 2.0 wt%, bwc, referring to the suspensions. Another example is the use of HPEG-PCE with 45 EO units [108]. The activation of mortars, based on a PC 52.5 employing w/c = 0.50, were obtained with dosages of 0.3 and 0.6 wt%, bwc, referring to the solid content. Moreover, these composite seeds were used in a subsequent work to activate the hydration of a PC-metakaolin blend with increasing amounts of seeds [109]. In addition to the well-established pore refinement by MK, further pore refinement by the C-S-H seeding was reported based on MIP data. Unfortunately, the consequences for the durability performances have not been established so far. Two other works also described the stabilisation with HPEG-PCE superplasticizers and the PC hydration activation [110,111]. Finally, the stabilization with a MPEG-PCE type superplasticizer has also been reported [112]. In this case, the activation of mortars was based on a PC 42.5, with variable w/c values, 0.50, 0.40, and 0.35, and dosages of 0.5 and 1.0 bwc, respectively. The increase in compressive strengths was complemented with early age porosity values derived from MIP data.

C-S-H/PCE composite seeds (named in the publication: new multifunctional admixture) were mechanochemically prepared by the full hydration of PC 52.5 with excess water in the presence of a PCE superplasticizer. The reaction was conducted at 60 ºC for 48 h in a steel reactor equipped with a reflux condenser and a high shear stirrer. Finally, Na_2_CO_3_ was added to the suspension and the mixture was stirred for another 60 min. The final stabilised suspension had a pH of 13.1 [113]. The activation performances of this admixture were successfully tested in six low-carbon binders: (i) CEM III/A 32.5R, (ii) CEM III/B 42.5 N, (iii) CEM IV/A (V) 42.5R, (iv) CEM IV/B (P) 32.5R, (v) CEM I + FA, and (vi) LC^3^. The compositions of every binder and the relative percentage increase in compressive strength of mortars prepared with 1.25% bwc at 17 h, and 1 d, 7 d and 28 d, respectively, were detailed in the original publication. Not only secondary nucleation, but also sustained alkaline activation, were considered key for justifying the good strength promoting features of this admixture.

To end this subsection, C-S-H/PCE composite seeds were also prepared by the hydration of a mixture GGBFS and carbide slag (CS) after wet grinding [114]. The PC 52.5 activation performances of the resulting powders were characterised by a number of techniques including MIP.

#### 5.2.2. Stabilised Metal-Substituted C-S-H Seed Composites

In addition to preparing C-S-H nanoparticles with variable Ca/Si ratios, an extra choice is to incorporate different amounts of aluminium within the gel. This study was conducted and a C-A-S-H/PCE suspension with a moderate Al/Si ratio, i.e., 0.05, and Ca/Si = 1.0 showed the strongest seeding effect [115,116]. The cement hydration activation seems to be complex as it depends upon the element composition, effective particle sizes in the suspensions, and their morphology.

It has also been recently reported that C-S-H/PCE nanocomposites incorporating magnesium ions, especially at the Ca/Si ratio of 0.8 and Mg/Si ratio of 0.1, have better stabilities [117].

#### 5.2.3. C-S-H Coated Nanomaterial Seed Composites

Two types of nanomaterials have been coated with C-S-H through surface deposition. First, C-S-H was deposited on FA nanoparticles without the need to use polymer stabilisers [118]. Second, C-S-H was coated on calcium carbonate nanoparticles and the resulting composite suspension was stabilised by a PCE dispersant [119]. The accelerating performances of these seeds were evaluated in mortars derived from a PC 42.5 with a w/b = 0.40. The strengths at 1 day were significantly improved, but at 28 days were not.

### 5.3. Crystalline Calcium Silicate Hydrate Phases as Nucleation Seeds

There are some reports dealing with related calcium silicate hydrate minerals as nucleation seed agents. Afwillite, Ca_3_[SiO_4_][SiO_2_(OH)_2_]·2H_2_O, is known to accelerate the hydration of C_3_S pastes [42,120]. However, it is also known that it does not significantly enhance the hydration of PCs [42]. Xonotlite, Ca_6_[Si_6_O_17_][OH]_2_, and hillebrandite, Ca_2_SiO_3_[OH]_2_, were recently studied as model compounds for PC hydration activation [121]. Very poor performances were reported.

## 6. Activation of Cement Hydration by C-S-H Nucleation Seeding

### 6.1. Acceleration of Cement Hydration at Early Ages

It is well established that C-S-H nucleation seeding accelerates cement hydration at early ages, i.e., during the first hours and days. This is illustrated in Figure 2, which displays the calorimetric traces for two pastes (80 wt% of CEM I 32.5 R and 20 wt% of FA), unseeded and seeded with 4 wt% of Master X-Seed 100 (≈0.90 wt% referring to the solid content), from [67]. The overall binder hydration acceleration/enhancement can be noticed by five features highlighted in Figure 2. (1) For the seeded paste, 20FA4NA, the induction period is shorter, i.e., 1.3 h instead of 3 h (see red lines). (2) 20FA4NA shows a higher slope in the acceleration stage (pink lines). (3) It also displays a larger value of the heat-flow maximum for the main hydration (alite-due) peak (brown lines), which occurs earlier. (4) There is an enhancement of the second peak in the heat flow trace (aluminates-related). (5) Finally, there is a narrowing of the two larger peaks in the heat flow traces, see pink arrow. This narrowing, i.e., faster deceleration, is generally observed in C-S-H seeded pastes and its origin is not yet fully established. It likely arises from two contributions: (i) faster/enhanced aluminate dissolution (second peak) results in larger Al^3+^ species concentration in the pore solution, which is known to delay alite hydration [19,33,34,35,36]; and (ii) hydration acceleration consumes more water at very early ages, which may lead to slower hydration rates, after the main peaks, because of lower diffusion rates. Liquid water is both a reactant and the medium where all chemical (hydration) reactions take place.

Figure 3 displays a similar calorimetric study for blend cement pastes composed of 50 wt% of CEM I 52.5 R and 50 wt% of GGBFS from [80]. In this case, the pastes were seeded with 3 wt% of HyCon^®^ S, and increasing amounts of anhydrite. It can be seen that the four main accelerating signatures of C-S-H seeding are present in the traces, and they are not further elaborated on for this example. Now, we focus on the fifth feature, i.e., the faster deceleration in the seeded specimens. As expected, the seeded pastes show a faster deceleration after the main hydration peaks. Interestingly, this feature can be partly mitigated by the addition of anhydrite, see pink arrow in Figure 3. We interpret this observation as a clear support to the Al^3+^ delaying effect on alite hydration rate. Larger amounts of anhydrite may accelerate ettringite precipitation, and, therefore, less aluminium species are available in the pore solution (in equilibrium with the amount adsorbed on the C-S-H nanoparticle surfaces) which decreases the delays in alite hydration. This ultimately leads to a slight increase in the cumulative heats for the C-S-H seeded pastes with increasing amounts of anhydrite, see Figure 3.

To investigate the possible role of the w/c ratios, Figure 4 shows a calorimetric study for a CEM I 42.5 R paste at two relevant w/c ratios (0.50 and 0.40), from [71,78]. In these works, the pastes were seeded with 2 wt% of different Master X-Seed admixtures (≈0.50 wt% referring to the solid contents). The four main accelerating signatures of C-S-H nucleation seeding can be readily observed. Quantitatively, the effects are smaller in plain PCs than in blended cements with SCMs, which encourages the future use of C-S-H seeding for accelerating the hydration of low-carbon cements. Figure 4 also shows the typical (undesirable) fifth feature of faster cement deceleration, after the enhancements of the alite and aluminate hydration peaks. Interestingly, XS130 admixture accelerated the aluminate peak the most, and also displayed the largest deceleration feature. Moreover, for the same cement and with the same C-S-H seeding dosage, the paste containing STE53 admixture and w/c = 0.40, see Figure 4c, has a smaller deceleration rate than XS130 with w/c = 0.50 (in the period 18–24 h), see Figure 4a. This key observation indicates that (further) water consumption, i.e., very much related to the mass diffusion, is not the main contributing factor for the observed enhanced deceleration. The analysis of the reported behaviours indicates that larger aluminate species content in the pore solution after C-S-H seeding is likely the main contributing factor to the faster deceleration in the C-S-H seeded binders.

To the best of the authors knowledge, so far there are only two papers reporting in situ RQPA results for C-S-H seeded samples during the first day of hydration [71,80]. From LXRPD data, Li et al. [80] reported faster dissolution rates of C_3_S, C_3_A, and anhydrite for the seeded PC-GGBFS blends. However, the same study reported a decrease in ettringite content after seeding. Moreover, for the 50-50 blend, the CH content was reported to increase, at 24 h, from 6 wt% for the unseeded paste to 14 wt% for the 3 wt% C-S-H seeded one. This very large increase in portlandite cannot be justified neither by the increased reactivity of alite nor by the enhanced hydration of the slag.

In situ SXRPD measurements of early age cement hydration [122,123] were conducted in transmission within rotating capillaries, and they are free from the experimental errors typical of LXRPD working in reflection with samples covered by a polymer foil (the three main ones being: water microbleeding, CH preferred orientation, vertical particle segregations before setting). SXRPD has been very recently applied to the hydration of C-S-H seeded cement pastes [71]. Figure 5 displays the time-evolution of the different crystalline phases from Rietveld analysis of the SXRPD data for CEM I 42.5 R pastes with w/c = 0.50. It can be seen that the alite reaction (dissolution/hydration) is slightly accelerated by C-S-H nucleation seeding (with different Master X-Seed admixtures, constant dosage at 2 wt% of the commercially available stabilised suspension), see Figure 5a. TIPA dosage at 0.05 wt% did not alter the alite and gypsum dissolution rates or portlandite crystallization rate. Interestingly, C-S-H seeding clearly enhanced gypsum dissolution, see Figure 5c. Moreover, C-S-H seeding also enhanced C_3_A and C_4_AF dissolutions, mainly after 10 h. As expected, an enhanced gypsum and C_3_A dissolutions, mainly for XS130 admixture, resulted in larger amounts of crystallised ettringite, see right-axis of Figure 5c. For the seeded paste with XS130 admixture, this study shows that after 12 h the C_3_A dissolution is enhanced, and the ettringite crystallization is also increased. This readily explains the ‘second’ aluminate peak measured in the calorimetric study which took place at 15 h, see blue trace in Figure 4a. Finally, a careful observation of Figure 5a for XS130 seeded paste, shows that the alite dissolution rate decreases at about 12 h which justifies the narrowing of the calorimetric trace after the aluminate peak. This decrease in overall reaction rate is due to a sluggish reaction/hydration of the alite likely because of the additional aluminium species content in the pore solution.

The acceleration of the pastes by C-S-H seeding is generally characterised by calorimetry, for an overall understanding, and by RQPA of powder diffraction data, i.e., quantitative phase evolutions, for a more detailed mechanistic knowledge. The acceleration of mortars is generally studied by mechanical strength measurements which are sometimes complemented by shrinkage assays. An additional way that the acceleration can be measured is to use UPV which has been reported in seeded pastes [63] and seeded mortars [71]. Figure 6 displays the C-S-H seeding acceleration features of BRC pastes by calorimetry and the corresponding mortars by UPV. By calorimetry, the induction period shortens, and the rate of the acceleration stage increases for the three Master X-Seed admixtures. Interestingly, XS130 also provokes a very sharp ‘aluminate’ peak at about 14 h which is followed by very slow hydration processes in the 18–30 h’s time range. This is fully consistent with the hypothesis of aluminate species being responsible for slower alite hydration rates. TIPA does not modify the first signatures and it only increases the aluminate peak. All these processes are mirrored in the corresponding mortars, see Figure 5c,d. The main hydration peak takes place in the unseeded paste at ≈20 h and occurs at ≈7 h for the mortar. Very interestingly, the sharp peak that takes place for XS130 seeded pastes at ≈16 h occurs at ≈13 h for the corresponding mortar. TIPA changes the first peak of the mortar only slightly, but it increases and accelerates the aluminate peaks. All signatures are consistent between pastes and mortars. This study showed that UPV can be trustily used to investigate the consequences of C-S-H nucleation seeding in mortars.

### 6.2. Improving Mechanical Strength Performances at Early and Later Ages

Here, we do not discuss the papers showing improved mechanical strength in the range 12 h to ≈2 days by C-S-H nucleation seeding. This can be derived from the publications gathered in Section 5. Most of these investigations reported large mechanical strength improvements before one day of hydration. Here, we are interested in comparing results at an early age (1–3 days) and the corresponding ones at later ages, i.e., 28 d. Moreover, low-temperature results are not listed in this subsection due to their dependence on pre-curing features, but these works were highlighted in Table 1.

Table 2 lists the compressive strength variations (relative values with respect to the corresponding unseeded specimens) at early and at late hydration ages. It is important to notice that the C-S-H dosage, referring to the solid content of the admixture, is quite variable. Table 2 gathers values spanning by more than an order of magnitude, from as low as 0.20 wt% to as high as 3.0 wt%. The most commonly used value is close to 0.5 wt% (referring to the solid content, bwb). In the opinion of the authors, many more systematic studies are needed to establish the optimum C-S-H nucleation seeding dosage that could be binder and application dependent.

A close inspection of the results shown in Table 2 reveals that, as expected, at early ages large improvements in the compressive strengths are generally observed after seeding. Improvements larger than 100% are commonly measured at 12 h (ideal for precast applications). Improvements close to 50% are generally observed at 1 day (important for fast formwork management). Concerning the strength results at 28 days by C-S-H nucleation seeding, there are scattered results, but generally there are improvements although quantitatively lower, close to 10–20%. Quantitatively, and at early ages, it seems that C-S-H seeding increases the compressive strengths of plain PCs more than in related blends with SCMs. However, at 28 d, the situation is reversed and the improvements in mechanical properties of blends are larger than in plain PCs. In this sense, the synergy of C-S-H nucleation seeding and alkanolamine addition (to enhance calcium aluminate hydration) seems to be a very promising strategy to have improved mechanical strengths at 28 days and later [71,77,78,82,86,87].

**Table 2 materials-16-01462-t002:** Selected works showing the effects of C-S-H nucleation seeding on compressive strengths at RT, reported as relative variation with respect to the unseeded samples. The variations, when available, are given at 12 h, and 1 d and 28 d.

Admixture ^$^	Sample	Dosage ^#^	Binder	w/b	Compr. Strengths Variation (%) ^@^			Ref
		(wt%)			12 h	1 d	28 d	
XS100	concretes	0.88	CEM I 32.5R	0.40	+70	+9	+14	[61]
XS100	concretes	0.44	80%CEM I 32.5R—20%FA	0.40	+100	+18	+3	[61]
XS100	concretes	0.44	CEM V/A with MS	0.30	+71	+36	+3	[62]
XS100	pastes	0.88	80%CEM I 32.5R—20%FA	0.30	+173	+72	+2	[64]
XS100	concretes	0.44	CEM I 42.5R	0.29	+26	+21	+12	[65]
XS100	mortars	0.44	aBRC 42.5	0.50	-	−2	+2	[71]
XS55	pastes	0.22	PC A3000	0.40	-	+23	+15	[40]
XS55	pastes	0.22	80%PC A3000—20%MK	0.40	-	+34	+8	[40]
XS55	pastes	0.22	60%PC A3000—40%MK	0.40	-	+28	+7	[40]
XS120	mortars	0.42	80%CEM I 42.5N—20%CC	0.50	-	≈+40	≈0	[68]
XS1500	concretes	0.23/0.35	40%PC—60%FA	≈0.25	-	≈+19/+8	≈+9/−3	[70]
XS1500	concretes	0.23/0.35	30%PC—70%FA	≈0.25	-	≈−6/+8	≈+4/−7	[70]
XS130	mortars	0.56	CEM I 42.5R	0.50	-	+57	+2	[71]
XS130	mortars	0.56	aBRC 42.5	0.50	-	+29	+17	[71]
XS130	mortars	0.56	BRC 42.5	0.50	-	+13	+13	[71]
XS130	mortars	0.56	aBRC-LC^3^-32	0.40	-	+29^2d^	+19	[77]
XS130	mortars	0.56	aBRC-LC^3^-48	0.40	-	+23^2d^	+18	[77]
XS130	mortars	0.56	CEM I 42.5R	0.40	-	+42	0	[78]
XS130	mortars	0.56	aBRC 42.5	0.40	-	+21	−11	[78]
STE53	mortars	0.56	CEM I 42.5R	0.50	-	+47	+17	[78]
STE53	mortars	0.56	CEM I 42.5R	0.40	-	+42	+9	[78]
STE53	mortars	0.56	aBRC 42.5	0.50	-	+41	+7	[78]
STE53	mortars	0.56	aBRC 42.5	0.40	-	0	+2	[78]
HyCon	mortars	2.0	CEM I 52.5N	0.30	-	0	≈−8	[79]
HyCon	mortars	2.0	CEM I 52.5N	0.35	-	≈+19	0	[79]
HyCon	mortars	1.5/3.0	50%CEM I—50%GGBFS	0.50	-	+110/+159	−2/+5	[80]
HyCon	mortars	1.5/3.0	25%CEM I—75%GGBFS	0.50	-	+179/+297	+7/+29	[80]
HyCon	mortars	1.5/3.0	05%CEM I—95%GGBFS	0.50	-	+106/+221	+50/+82	[80]
VIVID	mortars	0.5/1.0	P·I 42.5 & 2% EVA	0.50	-	+31/+41^3d^	+1/+2	[81]
VIVID	mortars	1.0/2.0	P·I 42.5 & 6% EVA	0.50	-	+49/+126^3d^	+11/+60	[81]
VIVID	pastes	0.21/0.42	P·II 42.5 & 0.05% TIPA	0.40	-	+8/+12^7d^	+12/+15	[84]
VIVID	pastes	0.21/0.42	P·II 42.5 & 0.10% TIPA	0.40	-	+8/+12^7d^	+6/+13	[84]
VIVID	mortars	0.21/0.42	P·I 42.5	0.50	-	+21/+53^3d^	+9/+20	[85]
VIVID	mortars	0.21/0.42	P·I 42.5 & 1% Na_2_SO_4_	0.50	-	+36/+55^3d^	+10/+21	[85]
VIVID	mortars	1.0/2.0	P·I 42.5 & 0.05% TIPA	0.50	+93/+159	+20/+68	+3/+5	[87]
VIVID	mortars	1.0/2.0	P·I 42.5 & 0.10% TIPA	0.50	+81/+150	+17/+60	+4/+2	[87]

^$^ The acronyms for these commercial admixtures correspond to the names given in the first column of Table 1. ^#^ The C-S-H seeding dosages refer to the solid content of the admixtures and calculated by us with the known compositions of the admixtures when not explicitly stated by the original authors. Hence, these numbers should be taken cautiously. @ The superscripts “2d”, “3d” and “7d” refer to 2, 3 and 7 days, respectively.

### 6.3. Improving Durability Performances of Binders at Later Ages

The enhanced durability properties of concretes by using nanoparticles-containing admixtures is known [26]. Moreover, there are many reports of C-S-H nucleation seeding showing lower porosities (both smaller overall porosities and narrower threshold pore entry sizes) by MIP, see, for instance Figure 7. However, despite these encouraging preliminary results, the effects on durability properties of C-S-H nucleation seeding have not been thoroughly investigated so far. It could be hypothesised that moving part of the C-S-H nucleation from the surfaces of alite to the pore solution, will result in more homogeneous pastes, and hence, with lower permeabilities to harmful species, such as CO_2_, Cl^−^ or SO_4_^2−^. To the best of the authors knowledge, so far there are no papers reporting comparisons of unseeded and seeded samples dealing with Cl^−^ and SO_4_^2−^ resistances. There was a report showing that the chloride binding capacity was improved by synergetic use of TIPA and C-S-H nucleation seeding [84]. This could result in improved chloride resistance, but this must be directly proved.

Next, the few reports dealing with direct durability characterization after C-S-H nucleation seeding are discussed. In 2018, it was already reported that 3 wt% C-S-H seeding reduced the water permeation front by 50% [41]. This result was mainly attributed to a refinement of the capillary pore network, with higher tortuosity values. This finding was obtained for mortars, and it has been independently extended to concretes where seeded samples were shown to have lower water absorption values [65]. Moreover, smaller water absorption values have also been reported for C-S-H seeded pastes cured at low temperatures with different pre-curing patterns [73]. Very interestingly, the carbonation resistances of concretes produced in three ways were recently compared [118]. The reference concrete had 15% replaced PC 42.5 with FA. Two C-S-H seeded concretes were prepared by using plain C-S-H seeds and C-S-H deposited on the surfaces of FA. The accelerated carbonation results showed that, after standard curing by 28 d, the carbonation depth decreased by ≈5% when comparing the reference concrete and the one seeded with 0.5 wt% of C-S-H. Moreover, the decrease in carbonation depth was slightly larger, ≈8%, when seeding with FA coated by C-S-H nanoparticles [118].

Finally, and importantly, frost resistance was notably improved by C-S-H nucleation seeding in a blend composed by 70 wt% of CEM I 52.5 R and 30 wt% of GGBFS [74]. A total of 0.56 wt% of seeding (referring to the solid content) with Master X-Seed 130 was enough to protect the pastes in the tests. The samples were subjected to a total of 120 rapid freezing and thawing cycles, with temperatures ranging from −18 °C to +4 °C. However, similar and larger C-S-H dosages (up to 1.4 wt%) were not able to protect the pastes composed by 50 wt% of the same PC and 50 wt% of GGBFS [74]. The enhanced frost resistance was mainly attributed to the accelerated pozzolanic reaction rate and to the increased amount of pore-filling hydration products.

The above reported durability results, if reproduced and expanded, may indicate that durability performances could be improved by C-S-H nucleation seeding. This behaviour arises from the lower porosities and higher tortuosities in the seeded binders. Clearly, more research is needed here.

## 7. Current Understanding of C-S-H Nucleation Seeding in Cements

The understanding of C-S-H nucleation seeding effects is advancing with the new reports. Obviously, the two main effects are maintained: the secondary nucleation process consequences and the pore solution content modification effects. However, it is becoming apparent that the situation is more complex than just alite hydration acceleration, due to the interplay of factors. On the one hand, alite hydration is physically accelerated at very early ages by the additional surfaces, effective filler effect. This is evidenced in the calorimetries as the signatures (1), (2), and (3) described previously. However, C-S-H seeding also accelerates the aluminates and sulphate dissolution rates which is directly seen in the calorimetries as signature (4). Due to a higher content of aluminate species in the pore solution, alite hydration slows down; slower hydration rates could take place approximately after 1 day. Moreover, because the aluminate peak is very sensitive to the sulphate content, the sulphate availability also performs an indirect role in the C-S-H seeding features. In any case, C-S-H seeding at very early ages promotes two connecting anhydrous particle processes: C-S-H precipitation and ettringite crystallization. This is at the heart of the mechanical strength improvements at 12 h and 1 d. Afterward, other factors (sulphates and aluminates) perform important roles.

On the other hand, C-S-H nucleation seeding, for the same overall degree of hydration, seems to change the balance between inner and outer C-S-H gels. Generally, less dense outer C-S-H gel has comparatively larger volume filling characteristics, which yields better mechanical properties and lower porosities. Hence, even at the same degree of alite hydration, the benefits of C-S-H seeding in cement performances are noteworthy because the same amount of newly formed C-S-H is better able to connect the anhydrous particles leading to higher compressive strength and lower porosities. C-S-H densities at the microscale can span from lower than 1.80 g/cc (outer C-S-H with large content of small gel pore water) to higher than 2.10 g/cc (inner C-S-H with small amount of gel pore water and higher C/S ratio), i.e., up to a 15% difference. Our current understanding of the C-S-H nucleation seeding implications for PC hydration is shown in Figure 8 and detailed in the figure caption. The general variations of the different components, with C-S-H seeding, are highlighted in the right part of that figure. These features are not expected to significantly change for SCMs-containing blends, but more research is needed to understand the fine details.

## 8. Conclusions and Future Research Need

It is well-established that calcium silicate hydrate (C-S-H) nucleation seeding accelerates cement hydration at early ages, resulting in higher mechanical strengths. In addition, denser microstructures are generally observed at 28 days or later which often exhibit slightly improved mechanical properties. Furthermore, a synergistic effect with alkanolamines is being reported, which can improve mechanical strength performances at early and later ages. It has also been observed that care is not always taken in reporting the experimental details of C-S-H seeding. Here, it is strongly recommended to report the employed C-S-H seed contents referring to the solid content, in order to link the results of different research groups and/or the usage of different admixtures. Moreover, not only the origin, but also the full brand description of the admixture should be indicated.

Concerning future investigations, there are several fields which clearly deserve more research. This analysis is mainly focused on the usage of commercial (stabilised) C-S-H nucleation seeding because of its large availability. First, more thorough studies are needed with variable C-S-H dosages. Currently, ≈0.50 wt% C-S-H dosage (referring to the solid content) is commonly employed in plain Portland cements. However, higher dosages could be needed for blends containing high amounts of supplementary cementitious materials. Several studies have reported a limit where no further improvements, as measured by compressive strengths, are obtained. This limit, ranging 1.0–2.0 wt% for Portland cements, is not known for Portland cement-supplementary cementitious materials blends, and it could be binder and application dependent. Second, the synergistic effect of C-S-H seeding and alkanolamine addition should be further investigated. It seems that higher mechanical strengths can be obtained and also denser binders. Third, C-S-H seeding for accelerating these blends should be widely studied as it can contribute to addressing one of their main challenges, the poor mechanical strengths at early ages. This approach could have the beneficial side-effect of increasing the carbonation resistance of the final binders. Fourth, systematic investigations addressing variable sulphate and aluminate contents of the binders are needed. Here, it is remarked that, as discussed in the main text, the aluminate and sulphate contents impact the alite hydration rate. The optimum sulphate content is not only binder-dependent, but it could also be a function of the admixtures to be employed. Last, many more durability studies should be performed comparing unseeded and C-S-H seeded binders.

## Figures and Tables

**Figure 1 materials-16-01462-f001:**
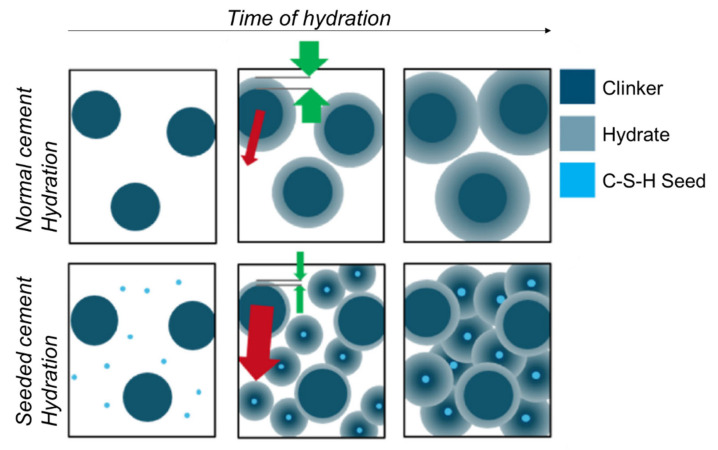
Schematic representation of the role of C-S-H nucleation seeding on the hydration of PC. The green arrows highlight the movement of the C-S-H growth away from the dissolving clinker particle volumes (inner C-S-H) towards the capillary porosity regions (outer C-S-H) caused by the nucleation seeding. The red arrows note the concentration gradients in ionic species which are exacerbated by C-S-H nucleation seeding. Reproduced from reference [25] with permission from Elsevier.

**Figure 2 materials-16-01462-f002:**
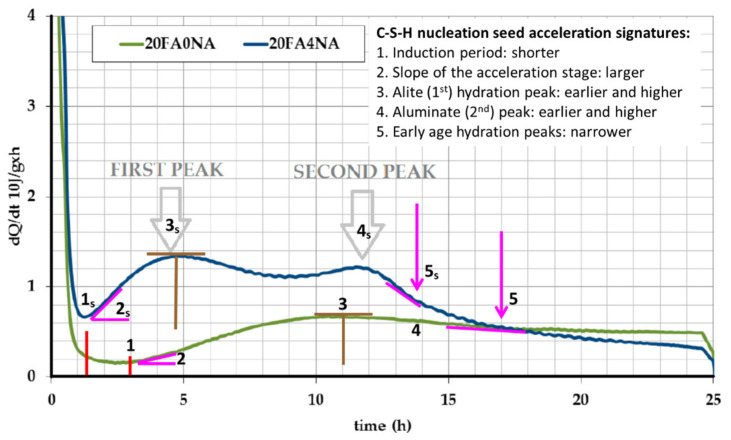
Calorimetric curves for a paste of CEM I 32.5 R substituted by 20 wt% of FA, w/b = 0.30 (green trace, 20FA0NA), and the same paste with a 4.0 wt% addition of XS100, amount referring to the admixture (blue trace, 20FA4NA). For the meaning of the lines and arrows, see the text. The subscript ‘s’ refers to the signatures in the seeded pastes. Adapted from reference [67] with permission.

**Figure 3 materials-16-01462-f003:**
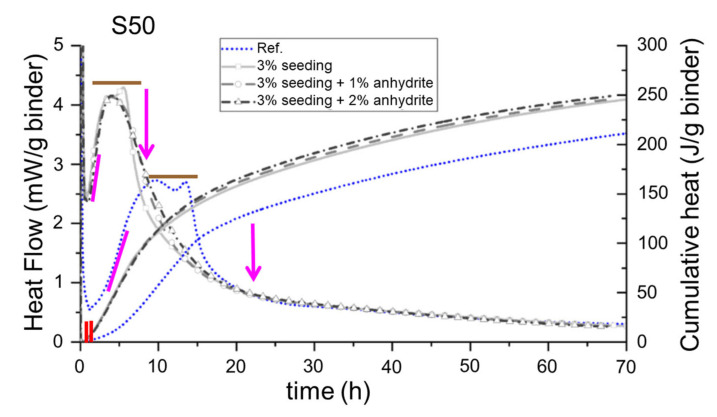
Calorimetric traces for pastes based on CEM I 52.5 R substituted by 50 wt% of GGBFS, w/b = 0.40. The trace for the reference paste (unseeded) is blue. The seeded pastes, 3 wt% of C-S-H stabilised nanoparticles, have increasing contents of anhydrite (grey traces). Symbols (lines and arrows) as in Figure 2. Adapted from reference [80] with permission from Elsevier.

**Figure 4 materials-16-01462-f004:**
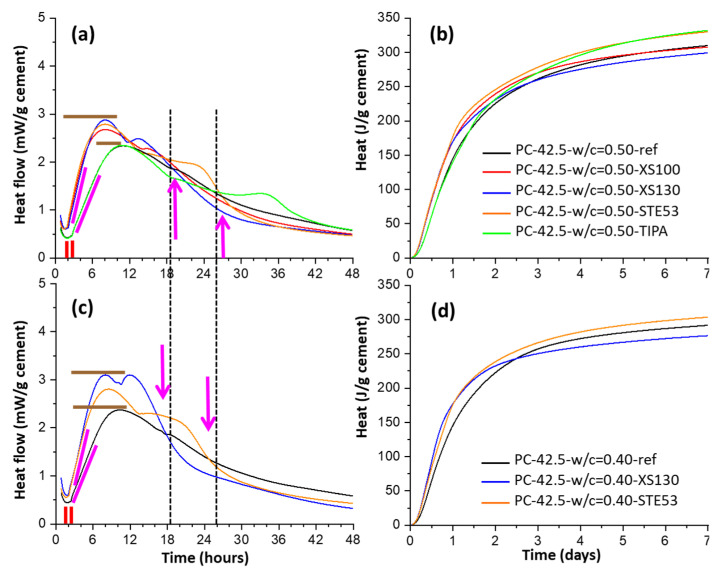
Calorimetric traces for pastes of CEM I 42.5 R with w/c = 0.50 and 0.40, seeded with 2 wt% of different Master X-Seed admixtures. Data for a paste with 0.05 wt% of TIPA is also given as a second reference. (**a**) Heat flow traces, w/c = 0.50, (**b**) cumulative heat data, w/c = 0.50, (**c**) heat flow, w/c = 0.40, and (**d**) cumulative heat, w/c = 0.50. Symbols (lines and arrows) as in Figure 2. Replotted from data of the authors of this work originally reported in references [71,78].

**Figure 5 materials-16-01462-f005:**
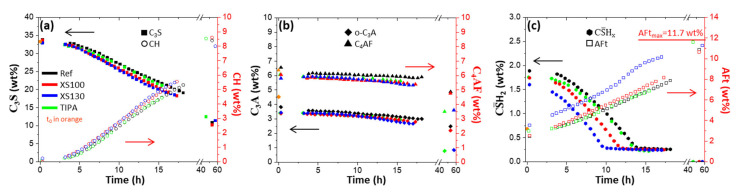
Rietveld quantitative phase analysis results for CEM I 42.5 R pastes, w/c = 0.40, seeded with 2 wt% of different Master X-Seed admixtures and 0.05 wt% of TPA. (**a**) C_3_S and CH contents with time, (**b**) C_3_A and C_4_AF contents with time, and (**c**) gypsum and ettringite contents with time. Replotted from data of the authors of this work originally reported in reference [71].

**Figure 6 materials-16-01462-f006:**
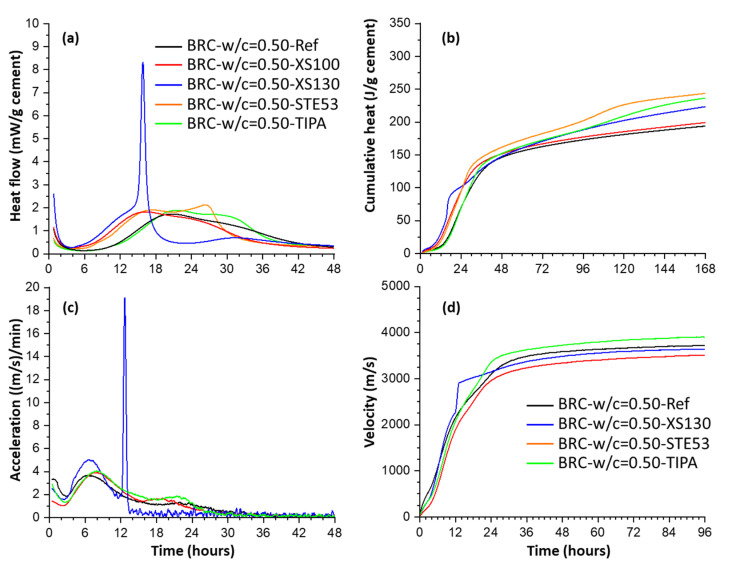
(**Top**) Calorimetric study for BRC pastes, w/c = 0.40, seeded with 2 wt% of different Master X-Seed admixtures and 0.05 wt% of TIPA. (**a**) Heat flow traces. (**b**) Cumulative heat values. (**Bottom**) UPV study for the same materials but processed as mortars. (**c**) Acceleration values, derivatives of the velocities. (**d**) Velocities through the mortars. Replotted from data of the authors of this work originally reported in reference [71].

**Figure 7 materials-16-01462-f007:**
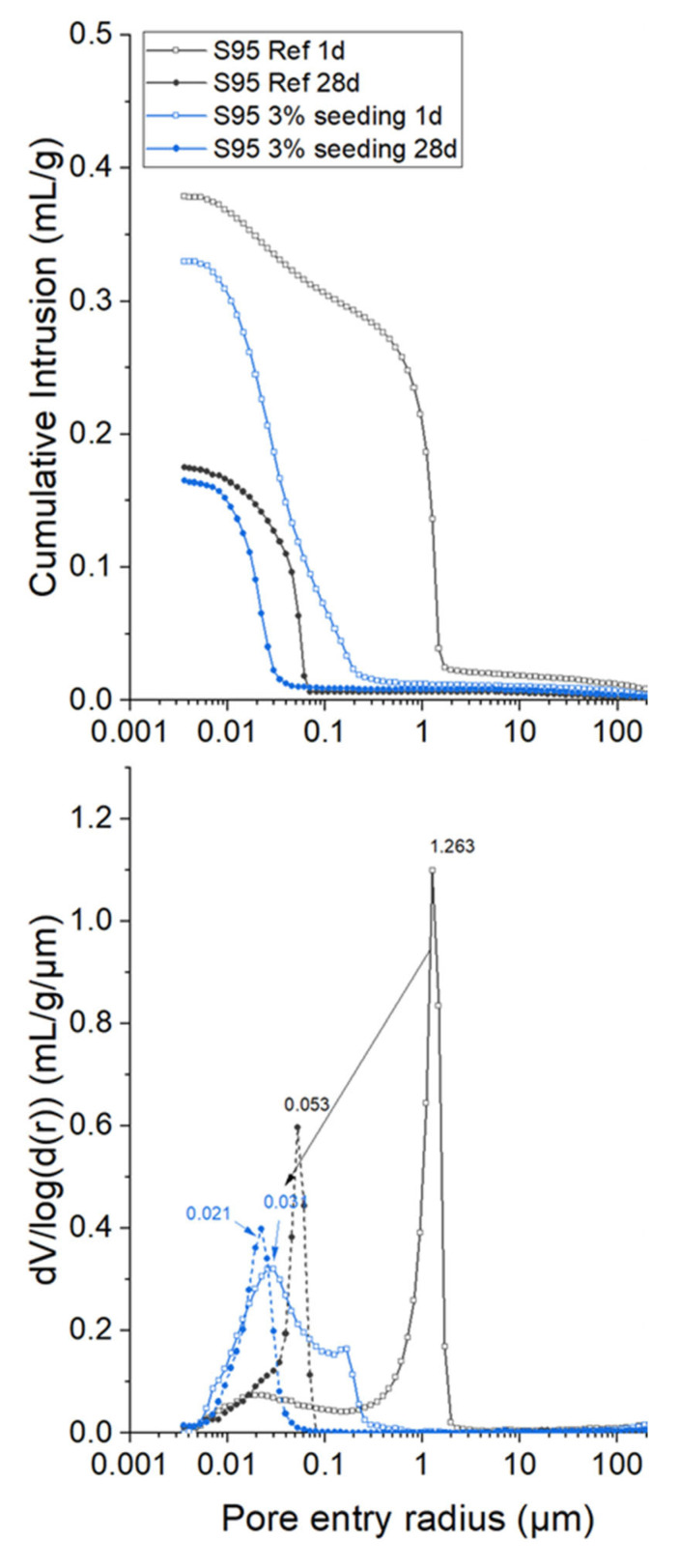
MIP data for a low-carbon cement blend (5 wt% PC 52.5 and 95 wt% GGBFS) with and without C-S-H seeding (HyCon^®^ S 7042 F) at 1 d and 28 days of hydration. Lower overall porosities and pore threshold entry sizes are clearly observed after seeding. Reproduced from reference [80] with permission from Elsevier.

**Figure 8 materials-16-01462-f008:**
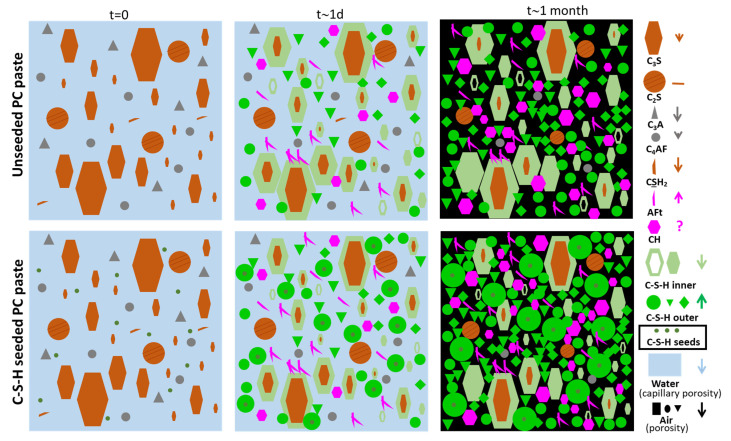
Schematic representation of the role of C-S-H nucleation seeding (dark green tiny particles) on the hydration of Portland cements (brown and grey particles) as currently understood by the authors. The different components are labelled in the right part. In addition to the previously known main features: (i) enhanced secondary nucleation in the pore solution space of outer C-S-H gel; and (ii) increase in the amount of (slightly lower density) outer C-S-H gel at the expenses of (slightly higher density) inner C-S-H gel; some additional properties should be noted: (iii) enhancement of calcium sulphate dissolution rate at early ages; (iv) slightly faster calcium aluminate dissolution rate(s) at early ages; (v) slightly faster ettringite crystallisation rate; (vi) smaller threshold pore size at later hydration ages; and (vii) in many cases, smaller overall porosities at later hydration ages. Note 1: the size of inner C-S-H gel at 1 day is not to scale but enlarged for better visualization. Note 2: the dissolution rates of the calcium aluminate phase (and therefore the crystallisation rate of ettringite) is further (synergistically) enhanced by the simultaneous addition of alkanolamines. This could further enhance alite hydration at early ages as C_4_AF intergrown with alite slows down alite hydration. Note 3: the faster dissolution rate of aluminates is very important for activating Al-containing SCMs, i.e., for instance in LC^3^ binders.

## Data Availability

There are not new raw data from this research.

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
