# Peer review of "Recent Advances in C-S-H Nucleation Seeding for Improving Cement Performances"

_materials, 2023, doi:10.3390/ma16041462_

Round 1

Reviewer 1 Report

The review paper entitled “Recent advances in C-S-H nucleation seeding for improving cement performances” was submitted on Materials journal for possible publication, the authors present a good contribution to admixtures of cement. The paper delivers the review of C-S-H seeding for improving cement performances. There are few comments may improve the paper before the publication.

-- Language needs to be polished.

-- Avoid abbreviations in Abstract and Conclusion parts.

-- Clustered references are observed in several places. Try to avoid such clustering differences, e.g., [18-22]. Readers could not be able to get the major findings from the literature. Keep the most relevant literature alone. In addition, the Introduction part need to be improved, some recent literatures on emission reduction of concrete and C-S-H seeds have not been included and cited, such as https://doi.org/10.1016/j.cemconcomp.2021.104319; https://doi.org/10.1016/j.apenergy.2022.120375; https://doi.org/10.1016/j.solener.2018.08.085; https://doi.org/10.1016/j.est.2020.101599, etc.

-- Some figures seem to be vague, including figure 1, 3 and 7. These figures need to be replaced by clearer version.

Author Response

Reviewer #1.

The review paper entitled “Recent advances in C-S-H nucleation seeding for improving cement performances” was submitted on Materials journal for possible publication, the authors present a good contribution to admixtures of cement. The paper delivers the review of C-S-H seeding for improving cement performances. There are few comments may improve the paper before the publication.

1.1. Language needs to be polished.

Answer: We have corrected the typo and errors that we have detected in the manuscript. Moreover, an English native scientist has edited the manuscript and she have corrected many minor mistakes/typos. All modifications are highlighted in blue in the revised version of the manuscript. They are too many to be given here.

1.2. Avoid abbreviations in Abstract and Conclusion parts.

Answer: This suggestion has been followed. No abbreviations neither in abstract nor in conclusion sections is kept. However, C-S-H is used seven times in conclusions. Therefore, we have taken the liberty to define it the first time and to keep, this well-established, wording in conclusions.

1.3 Clustered references are observed in several places. Try to avoid such clustering differences, e.g., [18-22]. Readers could not be able to get the major findings from the literature. Keep the most relevant literature alone. In addition, the Introduction part need to be improved, some recent literatures on emission reduction of concrete and C-S-H seeds have not been included and cited, such as https://doi.org/10.1016/j.cemconcomp.2021.104319; https://doi.org/10.1016/j.apenergy.2022.120375; https://doi.org/10.1016/j.solener.2018.08.085; https://doi.org/10.1016/j.est.2020.101599, etc.

Answer-1: We agree. Hence, we have separated some relevant references in the introduction section in order to give specific details about each of them. All modifications are highlighted in blue in the revised version.

Answer-2: We acknowledge that emission reduction is not covered here. However, we note that this is a vast field and we cannot cover it. Moreover, I am co-editor of Materials (and two other Journals) and I follow very strict rules in relation to citations. An analysis of the explicitly above-quoted papers by this reviewer follows:

https://doi.org/10.1016/j.cemconcomp.2021.104319 is dedicated to C-S-H seeding and it was cited as reference 73 in the manuscript.

https://doi.org/10.1016/j.apenergy.2022.120375 entitled “A clean strategy of concrete curing in cold climate: Solar thermal energy storage based on phase change material” is a very interesting work from Harbin but not related to seeding. Hence, we prefer not to reference it.

https://doi.org/10.1016/j.solener.2018.08.085 entitled “Development of calcium silicate-coated expanded clay based form-stable phase change materials for enhancing thermal and mechanical properties of cement-based composite” is a work from Harbin, devoted phase change materials and not to seeding. Hence, we prefer not to reference it.

https://doi.org/10.1016/j.est.2020.101599 entitled “Experimental research on an environment-friendly form-stable phase change material incorporating modified rice husk ash for thermal energy storage” is a work from Harbin, devoted phase change materials and not to seeding. Hence, we prefer not to reference it.

1.4 Some figures seem to be vague, including figure 1, 3 and 7. These figures need to be replaced by clearer version.

Answer: We agree, so, we have prepared clearer versions of figures 1, 3 and 7. We have used the high resolution web figures available from Elsevier. Moreover, the axes labels have been rewritten for clarity. However, it is noted that these figures are reproductions and hence, they are limited by the quality of the original figures.

Reviewer 2 Report

Please find the comments in the uploaded PDF file.

Author Response

This reviewer's feedback corresponds to manuscript reference materials-2074131, entitled "Nondestructive examination of carbon fiber-reinforced composites using
the eddy current method" and not to our manuscript, reference materials-2143952 entitled "Recent advances in C-S-H nucleation seeding for improving cement performances". Hence, it is not addressed.

Reviewer 3 Report

In the reviewed manuscript, recent advances in C-S-H nucleation seeding are well conducted and reviewed. I think the paper can be of interest to the readers of Materials. However, the results of this manuscript should be verified and confirmed. I suggest that the authors should have a minor revision and the suggestions are as follows:

- The abstract can state the innovative points of this study.

- The authors should explain the importance of C-S-H seeding in Introduction.

- The first column in Table 1 contains "??". Is it a typo? Please correct it.

- The lines in Figs. 3 and 7 are not clear, please correct the line.

- In Outlook, the authors suggest how researchers should approach the importance of C-S-H nucleation seeding and propose a reviewable protocol or standard experimental procedure.

Author Response

In the reviewed manuscript, recent advances in C-S-H nucleation seeding are well conducted and reviewed. I think the paper can be of interest to the readers of Materials. However, the results of this manuscript should be verified and confirmed. I suggest that the authors should have a minor revision and the suggestions are as follows:

2.1. The abstract can state the innovative points of this study.

Answer: We thank him/her for allowing us to strength the message of this review in the introduction. In order to do so, we have included the following statement in the abstract “It is noted that other features, in addition to the classic alite hydration acceleration, are covered here including the enhanced ettringite precipitation and the very efficient porosity refinement which take place in the seeded binders.”.

2.2 The authors should explain the importance of C-S-H seeding in Introduction.

Answer: We thank him/her for allowing us to clarify this. The following statement has been added at the end of the introduction. “The importance of C-S-H nucleation seeding is therefore due to its ability to accelerate the hydration of low-carbon cements at early ages without affecting their durability performances at later ages. This is reflected in higher mechanical strengths at early ages. Moreover, C-S-H seeding is also very important for concreting at low temperatures as hydration acceleration is required.”

2.3. The first column in Table 1 contains "??". Is it a typo? Please correct it.

Answer: This has been corrected. A footnote has been added to the table to clarify that the solid content reported value in the original publication, i.e. 0.134 wt%, is too low.

2.4 The lines in Figs. 3 and 7 are not clear, please correct the line.

Answer: This is the same comment than that of reviewer 1.4. Better figures have been prepared.

2.5 In Outlook, the authors suggest how researchers should approach the importance of C-S-H nucleation seeding and propose a reviewable protocol or standard experimental procedure.

Answer: Thanks, indeed.

Reviewer 4 Report

Paper ID: materials-2143952

Type: Review
Title: 
Recent advances in C-S-H nucleation seeding for improving cement performances

Authors: Ana Cuesta , Alejandro Morales-Cantero , Angeles G. De la Torre , Miguel A. G. Aranda

This study reviews recent advances in C-S-H nucleation seeding for improving cement performances. The paper is fair and can be considered for publication, although some issues need to be addressed. Following are some comments for the authors to consider: 

  1. Novelty in comparison to recent literature? Need to be emphasized.
  2. Please add a section (Review objective and methodology)
  3. Please add a section (Data collection and categorizing). The number of studies reviewed in this study and source type (journal, book, conference, others) may be presented.
  4. The last section name can be “Conclusion and future research need”.
  5. Throughout the text, some typos must be eliminated.

Author Response

This study reviews recent advances in C-S-H nucleation seeding for improving cement performances. The paper is fair and can be considered for publication, although some issues need to be addressed. Following are some comments for the authors to consider: 

3.1. Novelty in comparison to recent literature? Need to be emphasized.

Answer: Thanks, this makes a lot of sense. A novelty statement has been added at the end of the second section and it reads: “The novelty of this work resides on the discussion of the consequences of C-S-H nucleation seeding beyond the accelerating effect on alite hydration at very early ages. Specifically, several other outcomes are emerging from the analysis of the reports in recent years, including: i) the importance of faster ettringite crystallisation; ii) the possible slower alite dissolution rate after approximately one day because the higher con-centration of aluminate species in the pore solution; and chiefly iii) the lower porosities and pore entry thresh-old values of the seeded pastes.”.

3.2. Please add a section (Review objective and methodology)

Answer: Thanks. We have added such section with its corresponding content. This is reproduced here: “The main objective of the present review is to gather the latest data concerning the consequences of C-S-H nucleation seeding using commercial admixtures. This is done with the intention to develop a better mechanistic understanding that could results in further improvements. It must be noted that a wide range of techniques are being currently used to characterise the seeded products, and hence, a better mechanistic understanding can now be developed. C-S-H nucleation seeding is especially suitable for low-carbon binders and for concreting at low temperatures. In these cases, the slow hydration rates must be boosted to have competitive mechanical performances. Moreover, because denser binders are produced, all works related to durability characterization are discussed.

The employed methodology and the structure of the review is presented next. On the one hand, two tables gather the latest works using commercial admixtures, Table 1, and the relative improvement of the mechanical performances at early and later ages, Table 2. On the other hand, the structure of this work is detailed. Section 4 reports the data collection features. Then, key section 5.1 thoroughly describes the works employing commercial admixtures. The corresponding works developing and using laboratory-prepared C-S-H seeds are analysed, in less detail, in subsection 5.2. Then, the current understanding of the processes involved in the activation of cement hydration by C-S-H nucleation seeding are reviewed, section 6. This is divided in three subsections: i) the acceleration of cement hydration at early ages; ii) the improvement of mechanical strengths at early and later ages; and iii) the improvement of durability performances at later ages. Finally, section 7 gives an overall picture, as currently understood, of the mechanism at stake in C-S-H nucleation seeding. The review ends with a section of conclusions, where also the most important future research needs are highlighted.”

3.3. Please add a section (Data collection and categorizing). The number of studies reviewed in this study and source type (journal, book, conference, others) may be presented.

Answer: Thanks. We have added such section with its corresponding content, reproduced next.

“In this section, we briefly discuss the data collection procedure and an analysis of the referenced works. Most of the investigations cited here are papers from journals, amounting 91% of the all references, i.e. 112 out of 123. It is noted that this work references four book chapters, five proceedings and two patents. The investigations have been mainly obtained from the three available databases: i) Google Scholar; ii) Scopus; and iii) Web of Science. The searches were based on keywords “C-S-H” or “CSH” and “seeds” or “seeding” or “nanoseeding”. Moreover, the introductions of the resulting investigations led to some works which were not identified in the initial searches.

As a review on C-S-H seeding was published in 2018 [25], a time analysis seems adequate. It is noted that 82, out of the 123 references gathered here, have been published in 2019-2023. Therefore, at least 67% of the references were not covered in the previous review. Moreover, it is remarked that in 2021, 32 papers where published dealing with C-S-H seeding. This is the last year we consider that it has been fully covered here, as the searches for this review were carried out in November 2022. The corresponding publications in 2017, 2018 and 2019 were 5, 7 and 9, respectively. This four-fold increase in four years shows the growing importance and momentum of ‘C-S-H seeding’.”.

3.4. The last section name can be “Conclusion and future research need”.

Answer: Thanks, modified.

3.5. Throughout the text, some typos must be eliminated.

Answer: This is the same comment than 1.1 and it was addressed there.

Round 2

Reviewer 1 Report

All my concerns were addressed well. In my opinion, the paper can be accepted in the present form.